# Overcoming Referential Ambiguity in language-guided goal-conditioned Reinforcement Learning

**Hugo Caselles-Dupré, Olivier Sigaud, Mohamed Chetouani**
Sorbonne Université, CNRS, Institut des Systèmes Intelligents et de Robotique (ISIR)
Paris, France
casellesdupre.hugo@gmail.com

## Abstract

Teaching an agent to perform new tasks using natural language can easily be hindered by ambiguities in interpretation. When a teacher provides an instruction to a learner about an object by referring to its features, the learner can misunderstand the teacher's intentions, for instance if the instruction ambiguously refer to features of the object, a phenomenon called *referential ambiguity*. We study how two concepts derived from cognitive sciences can help resolve those referential ambiguities: pedagogy (selecting the right instructions) and pragmatism (learning the preferences of the other agents using inductive reasoning). We apply those ideas to a teacher/learner setup with two artificial agents on a simulated robotic task (block-stacking). We show that these concepts improve sample efficiency for training the learner.

## 1 Introduction

Language is intrinsically contextual and often ambiguous [18]. Linguistic ambiguity is a quality of language that makes communication shorter but open to multiple interpretations. As a consequence, the receiver of the message may need some additional information to reliably decode its meaning. A notable source of ambiguity is referential ambiguity [1]: when a person linguistically refers to an object through some features, the receiver can misinterpret the object referred to for another that shares the same features. Usually, humans can resolve such ambiguities in two ways: precisely specifying which item they refer to, or using contextual information such as relative position to another object. We would like artificial agents to resolve such ambiguities in the same way and benefit from the improved communication efficiency.

In this paper, we address referential ambiguity with a teacher/learner setup with two artificial agents in multi-goal environments. Both agents are equipped with an action policy and an instruction policy with which they can respectively act in the environment and communicate about goals using language. We consider environments where there are objects with multiple features, and the agents can only refer to one feature of the object they want to designate. The teacher provides instructions to the learner, which has to infer the desired goal associated to the instruction.

To resolve referential ambiguities in this setup, we draw inspiration from ideas in developmental psychology [14]. We define 1) pedagogical teachers who wisely choose instructions to avoid ambiguities (as opposed to naive teachers who randomly pick any valid instruction) and 2) pragmatic learners who reason inductively to better understand the intentions behind the teacher's instructions (as opposed to literal learners that do not adapt to their teacher).

To validate that it has inferred the right goal, the learner reformulates the goal it inferred with another instruction, from which the teacher deduces a goal, and tells the learner if goal inference was correct. When it is so, the learner can pursue the goal and use Reinforcement Learning to update its action

36th Conference on Neural Information Processing Systems (NeurIPS 2022).

policy. With this approach, we show that pedagogy and pragmatism can help resolve referential ambiguities for goal inference from instructions. This improves communication between the teacher and the learner and results in more sample efficient learner training.

**Related work**

Our work is rooted in the Goal-Conditioned Reinforcement Learning (GCRL) literature [8], in which agents learn to pursue and reach different goals by conditioning their action policy with representations of these goals. More specifically, we study agents that can both set their own goals (i.e. autotelic agents [8]) and be helped by other agents (i.e. teachable agents [19]). Previous work has shown that linking language to action is a powerful approach to improve upon classic GCRL approaches that are only based on actions [2, 7]. In our case, instructions can encode for goals, an approach called instruction-following [5, 12]. Most of these works rely on using optimal teachers and classic reinforcement learning learners without different goals and without optimizing the teacher or the learner to improve their communication efficiency by adding pedagogy or pragmatism.

Additionally, a long line of work in linguistics, natural language processing, and cognitive science has studied pragmatics: how linguistic meaning is affected by context and communicative goals, which led to a variety of frameworks for pragmatic communication [17, 10, 9, 13, 1, 16]. In our work, we take inspiration from the pragmatics ideas in linguistics to provide a simple way to add pedagogy and pragmatism to instruction-following tasks.

## 2 Methods

We present the teacher and learner implementation, and their communication through instructions.

### 2.1 Bayesian Goal Inference from Instructions

We formally introduce the Bayesian Goal Inference (BGI) mechanism used to infer goals from a goal-conditioned instruction policy (GCIP). In this work, the agents use it to infer goals from instructions provided by other agents.

**Action policy and Instruction policy**     All our agents are equipped with a GCIP $\pi_{\mathcal{I}}(i|g)$, which outputs an instruction probability distribution $i$ given a goal $g$. Additionally, the learner is equipped with a goal-conditioned action policy $\pi_{\mathcal{A}}(a|s, g)$, which outputs an action probability distribution $a$ given a state $s$ and a goal $g$. The action policy $\pi_{\mathcal{A}}$ is similar to a regular GCRL policy, and is used by the learner to reach goals in the environment. The instruction policy $\pi_{\mathcal{I}}$ is used by the agent to communicate goals to the other agent. It generates a valid language instruction for the given goal, which can be transmitted to other agents. In our case, those instructions use features of the objects (e.g. color) to refer to them and goals involve moving those objects such as "Put the red block next to the blue one".

**Inferring the Goal from an Instruction**     The instruction policy outputs a probability distribution for all valid instruction for a given goal. An agent can use its own instruction policy to infer the goal from an instruction (refer to [6] for a detailed explanation of BGI). To infer a goal from an instruction, by using Bayes' rule we can derive $\mathbb{P}(G|i)$, the probability distribution over the goal space $G$ given the instruction: $\mathbb{P}(G|i) \propto \mathbb{P}(i|G) \cdot \mathbb{P}(G) = \pi_{\mathcal{I}}(i|G) \cdot \mathbb{P}(G)$.

For each goal $g$, the prior $\mathbb{P}(G)$ is uniform if not specified otherwise. Given $\mathbb{P}(G|i)$, an agent can infer the goal from an instruction by either taking the most probable goal, or sampling from the distribution. To perform this inference, the agent uses its own instruction policy.

### 2.2 Training the Naive/Pedagogical Teacher and Literal/Pragmatic Learner

**Defining the teachers**     The naive teacher has an instruction policy for which all valid instructions for a given goal are equally likely. By contrast, the pedagogical teacher only assigns a positive probability to an instruction given a goal if this instruction is not valid for all other goals. Between those two extremes, we define teachers with attribute preferences. For instance, a teacher with color

preference will more likely refer to the color feature of objects rather than other features such as texture.

**Training the Learner with Teacher's instructions**   The learners' task is to learn how to master all goals starting from their untrained action policy $\pi_{\mathcal{A}}$, following the goals instructed by the teacher. In order to understand the instructions and communicate, learners have also access to an instruction policy $\pi_{\mathcal{I}}$. This instruction policy is initialized by assigning the same probability to all valid instruction for a given goal.

Given $\pi_{\mathcal{A}}$ and $\pi_{\mathcal{I}}$, the literal learner's training loop is the following. The teacher samples a goal $g$ and provides the learner with an instruction $i$ for this goal using its instruction policy. The learner receives the instruction and uses BGI to infer the goal $\widehat{g}$ requested by the teacher. In order to validate that the learner inferred the right goal, it samples another instruction using the inferred goal $\widehat{g}$ and its instruction policy, and transmits this evaluative feedback $i_f$ to the teacher. The teacher infers the goal $g_f$ of this evaluative feedback using BGI. If the inferred goal does not match the original goal ($g_f \neq g$), then the teacher starts over with a new instruction. If the inferred goal matches the original goal ($g_f = g$), the learner attempts at reaching this goal, collects a trajectory and add it to its replay buffer. Every $n$ episodes, the action policy of the learner is updated using GCRL.

**Literal and pragmatic learners**   Pragmatic learners differ from literal learners by iteratively improving their instruction policy during training. If the teacher announces that the inferred goal from the learner's feedback does not match the intended goal, the pragmatic learner will modify its instruction policy by lowering the probability of selecting the feedback instruction given the inferred goal: $\mathbb{P}(i_f|\widehat{g}) = \pi_{\mathcal{I}}(i_f|\widehat{g}) - \gamma$, with $\gamma$ being an hyperparameter (set to 0.1 in our work). Then, the instruction policy is normalized back to a probability distribution. This way, the pragmatic learner is able to adjust its instruction policy to maximize communication efficiency with the teacher.

# 3   Experiments and results

**Environment: Fetch Block Stacking**   We use the same "Fetch block-stacking" (FBS) environment as in [6]. In the instance used here, there are three blocks in the environment with different colors and textures: a red plain block (block 1), a blue plain block (block 2) and a blue striped block (block 3). These color and texture attributes are the basis for instructions used by the agents to refer to goals. The goal space is composed of 3 goals: each goal represents two blocks being close to each other. In order to create instructions about these goals, the agents are allowed to refer to one attribute of each blocks that has to be placed next to another. For instance, "Put the striped block next to the red block" would be a valid instruction for the goal of making block 1 and block 3 close. From those conditions, referential ambiguities emerge: when referring to a blue block, is the agent referring to block 2 (blue and plain) or block 3 (blue and striped)? Our experiments aim to show that with pedagogy and pragmatism, one can avoid those ambiguities.

**Action policy implementation**   The training procedure and policy architecture of the naive teacher are taken from GANGSTR [3] which already implements Fetch Block Stacking. For architecture, training and hyperparameters details, please refer to [3]. The policy is a message passing graph neural network [11]. This action policy is trained using Soft Actor Critic (SAC) [15], a state-of-the-art RL algorithm, combined with Hindsight Experience Replay (HER) [4].

**Main results**   We experiment with all combinations of teachers and learners, and report the results in Fig. 1(left), where we plot the GRA as a function of the number of instructions given by the teacher. As a metric, we use the Goal Reaching Accuracy (GRA), which is the average success rate of an agent over the goal space. The results clearly show the benefits of using pedagogy and pragmatism in our experimental setup. Indeed, the naive teacher + literal learner combination performs the worst compared to all other approaches that either use pedagogy or pragmatism. On the left of Fig. 1, we can see that both pragmatism and pedagogy alone improves performance, but the combination of the two yields an even greater performance gain. In Fig. 1(right), we show experiments with teachers that are non-ambiguous if you know their preferences (shapes or colors). We can see that a naive learner cannot benefit from these teachers, as they are not able to learn their preferences. By contrast, the pragmatic learner uses inductive reasoning to fine-tune its own instruction policy and is able to

learn the preferences of the teacher, thereby improving drastically its performance with results close to the pedagogical+pragmatic combination.

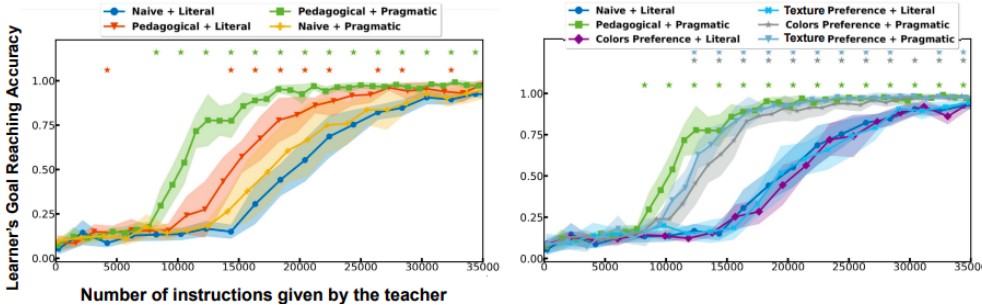

Fig. 1: Learner's GRA as a function of number of instructions given by the teacher for all possible combinations of teacher and learners. Left: results with pedagogy and pragmatism. Right: results with color/texture preference teachers.

**Why does the pedagogical teacher increase communication efficiency?** We provide a visual representation of the instruction policy of the pedagogical teacher in Fig. 2. This table presents all the 14 possible instructions and the three existing goals. To show the instruction policy, we insert in all cells the probabilities of picking the instruction given the goal.

We can clearly see that the pedagogical GCIP does not use the same instruction for two different goals, thus avoiding referential ambiguity. Besides, non-ambiguous instructions for a given goal are equally probable. Note that this not the only solution for a non-ambiguous GCIP, e.g. having only one non-ambiguous instruction for a given goal would work as well.

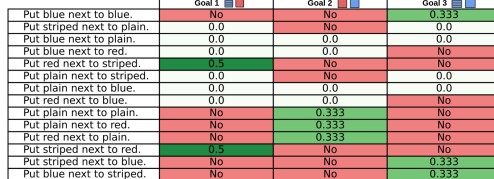

Fig. 2: Instruction policy of the pedagogical teacher. Green: instructions used by the pedagogical teacher, which are the only instructions that are not valid for other goals to avoid any type of ambiguity. Red: incompatibilities between the instruction and the goal. White: valid instructions with zero probability.

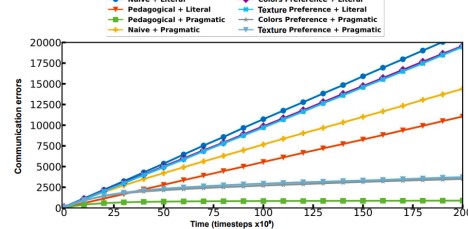

Fig. 3: Number of communication errors between the teacher and the learner. Pragmatic learners improve their instruction policy and reduce the communication errors wrt literal learners whose errors increase linearly.

**Why does the pragmatic learner increase communication efficiency?** In Fig. 3, we plot the number of communication errors as time progresses for all possible combination of teachers and learners. These errors stem from inference error from the learner or inference errors from the teacher when receiving the feedback. For literal learners, their instruction policy are fixed like the teachers, thus the communication efficiency between both agents remains the same during training. This is why in Fig. 3, the number of communication errors between the teacher and the learner increases linearly over time.

By contrast, the pragmatic learners improve their instruction policy over time. Indeed, the number of communication errors grows faster in the beginning of training, and then slows down. These results show that the learner communicates better and better, and thus requires less attempts to rightfully infer the goal of the instruction and get correct feedback from the teacher. For the pedagogical + pragmatic combination, we even see the number of communication errors becomes constant as the communication between the two agents becomes flawless.

**Conclusion**    In this paper, we have shown how referential ambiguities in instruction-following Reinforcement Learning can be countered by pedagogy and pragmatism ideas taken from the developmental psychology and cognitive sciences literature. In future work, we will explore how a teacher that can both provide instructions and demonstrations about goals can teach tasks efficiently to a learner. This will help understand the effects of the teaching signals on the training of the learner: when and how does a teacher should help the learner?

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
