# OpenReview forum: "Overcoming Referential Ambiguity in language-guided goal-conditioned Reinforcement Learning"
_NeurIPS.cc/2022/Workshop/LaReL — LaReL 2022_

### Official Review · Reviewer_nyAj · 2022-10-17
**Paper addresses some important issues but approach seems very simplistic and unscalable**

**Rating:** 5
**Confidence:** 4

**Review:**

Strengths

--Tackles an important problem setting: resolving ambiguity in language instructions.

--Writing is relatively clear throughout.

--Graphs clearly ablate the different approaches and show pedagogy and pragmatism help.

Weaknesses

-- The proposed approach relies on having a fixed instruction-goal probability table (14 instructions and 3 goals). It is unclear whether this approach would work in scenarios when the set of all possible instructions and goals are not known ahead of time, or when instructions and goals can be compositional combinations of simpler instructions and goals (such as in long horizon tasks).

-- Both aspects of the proposed approach (using pedagogy and pragmatism) seem quite simplistic and unscalable to more complicated scenarios where there are free-form language instructions and a combinatorial number of possible goals. Specifically, the proposed “pragmatism” for the learner seems to mainly be hard-coding a probability decrease in incorrect instruction guesses, which is only possible when the space of all possible instructions is known ahead of time (not a reasonable assumption for open-vocabulary systems that interact with humans). The proposed “pedagogical” approach for the teacher seems to simply be to hard-code probabilities in the instruction-goal table to ensure that all of its emitted instructions are unambiguous. It is unclear how this approach would scale to more realistic scenarios and handle the compositional semantics of more complex instructions and goals.

-- This paper would be a lot stronger, for instance, if the teacher started out as naive (and the learner started out as non-pragmatic), and both were able to, based on their rewards and communication, improve their instruction policies to be close to that of a pedagogical teacher and pragmatic learner.

-- Paper does not describe how the instructions are represented. Language embeddings? Some one-hot mapping?

-- Could have more clearly specified the action space of the block-stacking environment.

-- Lines 89-97 were unclear/hard to understand. A diagram would have helped.

-- Nitpick on Line 120: SAC is referred to as “a state-of-the-art RL algorithm,” which is not currently true. The statement would be more accurate if it were phrased as “a state-of-the-art online RL algorithm.”

-- Unclear what the stars on the figure 1 plot are.

-- Experiments are limited to one simple block stacking domain, but no block stacking is actually involved, since the goals are only to put two blocks of certain colors/textures next to each other. What constitutes “close enough” to count as a success?

---

### Official Review · Reviewer_c4ND · 2022-10-18
**What's the relationship between models of pragmatics and theory of mind**

**Rating:** 7
**Confidence:** 4

**Review:**

The work proposes a Bayesian Goal Inference model for instruction following/giving settings.  The motivation is clear.

The work references in developmental psychology but there's a rich literature on (machine) theory of mind, referential and language coordination games, etc that bears lots of resemblance to what the authors are discussing/proposing here.  Particularly, if any of the models are extended such that an agent has a model of the other's behavior. This literature might be of interest to the authors and guide extensions of their proposed models.

My main question is why in such a constrained setting does the instructor need to provide orders of magnitude more instructions than there are in the entire environment.  Does this mean the student is exposed to a tremendous number of redundant scenarios before learning (even in the best of models)?

---

### Decision · Program_Chairs · 2022-10-20

Accept